# Adaptive State Observer for Robot Manipulators Diagnostics and Health Degree Assessment

**Sanlei Dang [1], Zhengmin Kong [2],\*** 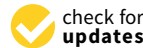**, Long Peng [1], Yilin Ji [1] and Yongwang Zhang [1]**

1   Metering Center of Guangdong Power Grid Co.Ltd., Guangzhou 510080, China;
    18588504175@gd.csg.cn (S.D.); 18825083646@gd.csg.cn (L.P.); 18802047836@gd.csg.cn (Y.J.);
    19928469628@gd.csg.cn (Y.Z.)
2   School of Electrical Engineering and Automation, Wuhan University, Wuhan 430072, China
\*   Correspondence: zmkong@whu.edu.cn; Tel.: +86-158-2716-6096

**Abstract:** To avoid serious damages caused by the dynamic environment, fault detection and health assessment are essential for an integrated robotic system. In this paper, we propose a fault detection algorithm and a health degree assessment approach for a robot manipulator system. Both the internal disturbance and the output measurement disturbance are considered in the proposed method. In addition, an adaptive observer is utilized to reconstruct the real system of robot manipulators. Under the proposed observer, the real system is estimated to detect the fault and obtain the health degree of the robot manipulator. The feasibility and reliability of the proposed fault detection algorithm and health degree assessment index for robot manipulator systems are proved by simulation experiments.

**Keywords:** fault diagnostics; health degree assessment; robot manipulator system; adaptive observer; equipment maintenance and management

## 1. Introduction

With the rapid development of computation and communication techniques, information science is increasingly becoming the research focus [1–3]. As an application of information science, robotics and intelligent systems are frequently utilized in industrial fields that require high safety, reliability and accuracy. Examples can be found in the advanced automated production and inspection line, mining, disposal of hazardous materials. However, robotics systems are always plagued by failures for many reasons such as incorrect operation and mechanical failure. Accurate fault detection and health assessment of robotics systems are significant, especially for robot manipulators [4–6].

For the past two decades, intensively investigations have been involved in fault detection of robot manipulators with different system models, see [7–13] and the references therein. So far, the fault detection for robot manipulators is still a key issue due to its great application potentials in both commercial and military areas [14–17].

Among the existing works related to fault detection for robot manipulators, the parameter estimation and the state observer are also two commonly used techniques. In [18], a model-based diagnostic scheme for actuator and sensor faults that may occur on a robot manipulator was present. The fault detection was achieved by a generalized observer scheme based on second-order sliding-mode approaches. However, the model-based detection scheme may suffer from modeling uncertainties due to various factors, such as aging equipments and internal system noise. To address it, Ma and Yang formulated a model-based actuator and sensor fault detection and isolation scheme for robot manipulators. They presented a nonlinear function to estimate the fault parameters with a pre-specified estimation error bound [19]. The parameter-variance model of a fault-tolerant

multi-sensor switching strategy for robot manipulators was considered in some recent studies [20–22]. A novel detection approach for multi-joint robots was proposed and a collision observer was designed iteratively through order reduction and re-selection of observer variables to improve computation efficiency [23]. Compared with the generalized state observers, the adaptive state observer shows a better performance in dealing with the disturbance [24]. By designing some adaptive nonlinear observers, a multiple fault detection scheme for robot manipulators was proposed in [25].

Despite the above mentioned literatures involved in fault detection with modeling uncertainties, the works addressing both internal and output measurement disturbances in robot manipulators diagnostics are rare. The various disturbances in the system make it a challenging work to accurately detect the fault manipulator. Motived by this, we consider the fault detection for robot manipulators with both internal and output measurement disturbances. A novel fault detection and health assessment approach is proposed to better reflect the possibility of robot manipulator failure. We design an adaptive observer to reconstruct the robot manipulator system, and the health degree of the manipulator is determined by comparing the real system with the original system. The feasibility of the proposed fault detection algorithm and the health assessment approach are verified by numerical simulations.

The remainder of this paper is organized as follows. We describe the kinetic model in Section 2. In Section 3, a fault detection approach for robot manipulators in disturbance condition is presented. The health assessment approach is given in Section 4, while the simulation results are provided in Section 5 for testifying the performance of our proposed approaches. Finally, the concluding remarks are provided in Section 6.

## 2. Kinetic Model

The dynamic model of an n-degree-of-freedom (n-DoF) robot manipulator [26] can be written in joint space as the following form

$$M(q)\ddot{q} + C(q,\dot{q})\dot{q} + G(q) - u^d = u^a, \tag{1}$$

where $q, \dot{q}, \ddot{q} \in R^n$ represent joint angular position, velocity and acceleration, respectively. $M(q) \in R^{n \times n}$ denotes the positive definite symmetric inertia matrix. $C(q,\dot{q}) \in R^{n \times n}$ and $G(q) \in R^n$ denote the Centripetal-Coriolis [27] and gravitational effect respectively. $u_d \in R^n$ is the external disturbance torque that obeys a Gaussian distribution with zero mean. $u^a \in R^n$ is the actual output torque generated by robot joint actuators.

Denote $u \in R^n$ as the expected output torque, then we have $u^a = u$ in the fault-free case. When the fault occurs, there is an error between the expected output torque and the actual output torque, which can be described as $\Delta u = u - u^a$. Let $x = [q, \dot{q}]^T$ be the state vector and $\theta_i = \frac{u_i^a}{u_i}$ be the ratio between the actual output torque and the expected output torque, we can establish a state-space model as

$$\begin{cases} \dot{x} = Ax + \Phi_0(x) + \Phi_1(x,u)\theta + w, \\ y = Cx + v, \end{cases} \tag{2}$$

where $A = \begin{bmatrix} 0 & I_n \\ 0 & 0 \end{bmatrix}$, $\Phi_0(x) = \begin{bmatrix} 0 \\ -M^{-1}(q)(C(q,\dot{q})\dot{q} + G(q)) \end{bmatrix}$, $\Phi_1(x,u) = \begin{bmatrix} 0 \\ -M^{-1}(q)U \end{bmatrix}$, $U = diag(u_1, u_2, \cdots, u_n)$, $\theta = [\theta_1, \theta_2, \cdots, \theta_n]^T$, $C = \begin{bmatrix} I_n & 0 \end{bmatrix}$. $w, v \in R^{2n}$ are the system noise and output noise signals that follow a Gaussian distribution with zero mean.

## 3. Fault Detection Algorithm

The objective of the fault detection algorithm is detecting relative errors between the expected output torque and the actual output torque by limited measurable output data [28]. With the increment

of relative errors, the fault possibility of the robot manipulator increases. In this context, we can transform the fault detection to an identification problem of parameter $\theta$. The relative errors between the expected output torque and the actual output torque can be represented as $|1 - \theta|$. Therefore, regarding $\theta$ as the unknown parameter, we apply the idea of adaptive state observer (ASO) [29] to identify the value of $\theta$. Before giving the fault detection algorithm, some assumptions are given below.

**Assumption 1.** *The state $x(t)$, the control $u(t)$ and the unknown parameters $\theta$ are bounded, i.e., $x(t) \in \mathcal{X}$, $u(t) \in \mathcal{U}$, and $\theta \in \mathcal{W}$ with compact sets $\mathcal{X} \in R^{2n}, \mathcal{U} \in R^n$ and $\mathcal{W} \in R^n$.*

**Assumption 2.** *The function $\Phi_0(x)$ and $\Phi_1(x)$ are Lipschitz with respect to $x$ uniformly in $u$, where $(u, x) \in \mathcal{U} \times \mathcal{X}$. Let the Lipschitz constants be $L_0$ and $L_1$, it can be obtained that*

$$\|\Phi_0(x) - \Phi_0(\hat{x})\| \leq L_0 \|x - \hat{x}\|, \tag{3}$$

$$\|\Phi_1(x) - \Phi_1(\hat{x})\| \leq L_1 \|x - \hat{x}\|. \tag{4}$$

*In order to identify the true value of $\theta$, we design an ASO as*

$$
\begin{cases}
\dot{\hat{x}}(t) = A\hat{x}(t) + \Phi_0(\hat{x}) + \Phi_1(\hat{x}, u)\hat{\theta}(t) - (S^{-1} + Y(t)P(t)Y^T(t))C^T(C\hat{x}(t) - y(t)), \\
\dot{\hat{\theta}} = -P(t)Y^T(t)C^T(C\hat{x}(t) - y(t)), \\
\dot{Y}(t) = (A - S^{-1}C^TC)Y^T(t) + \Phi_1(\hat{x}, u), \\
\dot{P}(t) = -P(t)Y^T(t)C^TCY(t) + kP(t),
\end{cases}
\tag{5}
$$

*where $Y(t) \in R^{2n \times n}, P(t) \in R^{n \times n}$ are two auxiliary matrices, the symmetric positive definite matrix $S$ is the unique solution of $kS + A^TS + SA - C^TC = 0$ with a constant $k$, $(A - S^{-1}C^TC)$ is the Hurwitz matrix, $U$ is bounded and $\|Y(t)\|$ has an upper bound.*

**Assumption 3.** *The input $U$ is such that for any trajectory $\hat{x}$ of system starting from $\dot{x}(0) \in X$, the matrix $CY(t)$ is persistently exciting in the following sense: $\exists \delta_1, \delta_2 > 0, \exists T > 0$,*

$$\delta_1 I_n \leq \int_t^{t+T} Y^T(\tau)C^TCY(\tau)d\tau \leq \delta_2 I_n, \quad \forall t \geq 0. \tag{6}$$

**Theorem 1.** *When the observation algorithm satisfies the Assumptions 1–3, there exists a constant $k_0$, such that for any $k > k_0$ and initial value $(x(0), \hat{x}(0)) \in X^2$, the mean of the estimated errors $E(\hat{x} - x)$ and $E(\hat{\theta} - \theta)$ converge to 0.*

**Proof.** Denote the errors between the true values of $x, \theta$ and the observed values of $\hat{x}, \hat{\theta}$ be $\tilde{x} = \hat{x} - x$ and $\tilde{\theta} = \hat{\theta} - \theta$, we can get the derivation of errors as

$$
\begin{cases}
\dot{\tilde{x}} = (A - S^{-1}C^TC)\tilde{x} + \tilde{\Phi}_0(\hat{x}, x) + \tilde{\Phi}_1(\hat{x}, x)\theta + \Phi_1(\hat{x}, u)\tilde{\theta} + Y(t)\dot{\tilde{\theta}} + (w(t) + S^{-1}C^Tv(t)), \\
\dot{\tilde{\theta}} = -P(t)Y^T(t)C^TC\tilde{x} + P(t)Y^T(t)C^Tv(t),
\end{cases}
\tag{7}
$$

where $\tilde{\Phi}_0(\hat{x}, x) = \Phi_0(\hat{x}) - \Phi_0(x)$ and $\tilde{\Phi}_1(\hat{x}, x) = \Phi_1(\hat{x}, u) - \Phi_1(x, u)$. Denote $\eta(t) = \tilde{x}(t) - Y(t)\tilde{\theta}(t)$, one can derive

$$
\begin{aligned}
\dot{\eta}(t) &= \dot{\tilde{x}} - \dot{Y}(t)\tilde{\theta} - Y(t)\dot{\tilde{\theta}} \\
&= (A - S^{-1}C^TC)\eta + \tilde{\Phi}_0(\hat{x}, x) + \tilde{\Phi}_1(\hat{x}, x)\theta + (w(t) + S^{-1}C^Tv(t)).
\end{aligned}
\tag{8}
$$

Firstly, we discuss the convergence of the system without noise signals, i.e., $w(t) = 0, v(t) = 0$. To avoid confusion, the variables $x, \theta, \eta$ in the noise-free system are denoted by $x_e, \theta_e, \eta_e$, respectively. Then, we can obtain

$$\begin{cases} \dot{\tilde{x}}_e = (A - S^{-1}C^TC)\tilde{x}_e + \check{\Phi}_0(\hat{x}_e, x_e) + \check{\Phi}_1(\hat{x}_e, x_e)\theta + \Phi_1(\hat{x}_e, u)\tilde{\theta}_e + Y(t)\dot{\tilde{\theta}}_e, \\ \dot{\tilde{\theta}}_e = -P(t)Y^T(t)C^TC\tilde{x}_e, \\ \dot{\eta}_e = (A - S^{-1}C^TC)\eta_e + \check{\Phi}_0(\hat{x}_e, x_e) + \check{\Phi}_1(\hat{x}_e, x_e)\theta. \end{cases} \tag{9}$$

Let $V_1(\eta_e(t)) = \eta_e{}^T(t)S\eta_e(t)$, $V_2(\theta_e(t)) = \theta_e{}^T(t)P^{-1}(t)\theta_e(t)$, where $S$ and $P(t)$ are given by Equation (5), we define the Lyapunov function $V(\eta_e(t), \theta_e(t)) = V_1(\eta_e(t)) + V_2(\theta_e(t))$. Based on Equation (8), it is derived that

$$\begin{aligned} \dot{V}_1(\eta_e(t)) &= \dot{\eta}_e{}^T(t)S\eta_e(t) + \eta_e{}^T(t)S\dot{\eta}_e(t) \\ &= \eta_e{}^T(A^TS + SA - 2C^TC)\eta_e + 2\eta_e{}^TS\check{\Phi}_0(\hat{x}_e, x_e) + 2\eta_e{}^TS\check{\Phi}_1(\hat{x}_e, x_e)\theta \\ &= -kV_1 - \eta_e{}^TC^TC\eta_e + 2\eta_e{}^TS\check{\Phi}_0(\hat{x}_e, x_e) + 2\eta_e{}^TS\check{\Phi}_1(\hat{x}_e, x_e)\theta. \end{aligned} \tag{10}$$

From the definition of $\eta_e$, it is clear that

$$\|\tilde{x}_e\| \leq \|\eta_e\| + \|Y(t)\| \|\tilde{\theta}_e\| \leq \|\eta_e\| + \gamma_m \|\tilde{\theta}_e\| \tag{11}$$

with $\gamma_m = \sup_{t \geq 0} \|Y(t)\|$. Based on Assumptions 1–3, it can be deduced that

$$2\eta_e{}^TS\Phi(\hat{x}_e, x_e) \leq 2 \|\eta_e{}^TS\| L_0 \|\tilde{x}_e\| \leq 2L_0(V_1 + \gamma_M \sqrt{\lambda_{\max}(S)\lambda_{\max}(P)}\sqrt{V_1}\sqrt{V_2}), \tag{12}$$

$$2\eta_e{}^TS\check{\Phi}_1(\hat{x}_e, x_e)\theta \leq 2 \|\eta_e{}^TS\| L_1\theta_M \|\tilde{x}_e\| \leq 2L_1\theta_M(V_1 + \gamma_M \sqrt{\lambda_{\max}(S)\lambda_{\max}(P)}\sqrt{V_1}\sqrt{V_2}), \tag{13}$$

where $\theta_M = \max_{1 \leq i \leq n} |\theta_i|$. Further, we set $c_1 = 2(L_0 + L_1\theta_M)$, $c_2 = c_1\gamma_M \sqrt{\lambda_{\max}(S)\lambda_{\max}(P)}$, then one can get

$$\dot{V}_1(\eta_e(t)) \leq -(k - c_1)V_1 + c_2\sqrt{V_1}\sqrt{V_2} - \eta_e{}^TC^TC\eta_e, \tag{14}$$

$$\begin{aligned} \dot{V}_2(\tilde{\theta}_e(t)) &= 2\tilde{\theta}_e{}^T(t)P^{-1}\dot{\tilde{\theta}}_e(t) - \tilde{\theta}_e{}^T(t)P^{-1}(t)\dot{P}(t)P^{-1}(t)\tilde{\theta}_e(t) \\ &= -kV_2 - \tilde{\theta}_e{}^T(t)Y^T(t)C^TCY(t)\tilde{\theta}_e(t) - 2\tilde{\theta}_e{}^T(t)Y^T(t)C^TCY(t)\eta_e{}^T. \end{aligned} \tag{15}$$

Hence,

$$\begin{aligned} V(\eta_e(t), \tilde{\theta}_e(t)) &= \dot{V}_1(\eta_e(t)) + \dot{V}_2(\tilde{\theta}_e(t)) \\ &= -(k - c_1)V_1 + c_2\sqrt{V_1}\sqrt{V_2} - kV_2 - (\tilde{\theta}_e{}^T(t)Y^T(t)C^TCY(t)\tilde{\theta}_e(t) + \\ &\quad 2\tilde{\theta}_e{}^T(t)Y^T(t)C^TCY(t)\eta_e{}^T + \eta_e{}^TC^TC\eta_e) \\ &= -(k - c_1)V_1 + c_2\sqrt{V_1}\sqrt{V_2} - kV_2 - x_e^T(t)C^TCx_e(t) \\ &\leq -(k - c_1 - \tfrac{c_2}{2})V. \end{aligned} \tag{16}$$

Obviously, we can find $k_0 = c_1 + \frac{c_2}{2}$, such that $V(t) \to 0$ when $k > k_0$. Thus, $V(t)$ converges to zero asymptotically, which implies that $\eta_e(t), \tilde{\theta}_e(t)$ are asymptotically stable. From $\tilde{x}_e(t) = \eta_e(t) - Y(t)\tilde{\theta}_e(t)$, it is known that $\tilde{x}_e(t)$ is also asymptotically stable. Therefore, we can get that the system is asymptotically stable. Since

$$\begin{cases} \dot{\tilde{x}}_e = (A - S^{-1}C^TC)\tilde{x}_e + \check{\Phi}_0(\hat{x}_e, x_e) + \check{\Phi}_1(\hat{x}_e, x_e)\theta + \Phi_1(\hat{x}_e, u)\tilde{\theta}_e + Y(t)\dot{\tilde{\theta}}_e, \\ \dot{\tilde{\theta}}_e = -P(t)Y^T(t)C^TC\tilde{x}_e, \end{cases} \tag{17}$$

it can be readily deduced that $(w(t) + S^{-1}C^Tv(t))$ and $P(t)Y^T(t)C^Tv(t)$ are bounded under the bounded noise signals $w(t)$ and $v(t)$. Thus, $\tilde{x}$ and $\tilde{\theta}$ are also bounded. That is to say, $\exists \varepsilon_1, \varepsilon_2 > 0$, $\|\tilde{x}\| < \varepsilon_1, \|\tilde{\theta}\| < \varepsilon_2$.

Besides, taking noise signals $w(t)$ and $v(t)$ into account, we can get

$$
\begin{cases}
\frac{d(E(\tilde{x}))}{dt} = (A - S^{-1}C^T C)E(\tilde{x}) + E(\Phi(\hat{x}, x)) + E(\check{\Phi}_1(\hat{x}, x))\theta + \Phi_1(\hat{x}, u)E(\tilde{\theta}) \\
\qquad\qquad + Y(t)E(\dot{\tilde{\theta}}) + (E(w(t)) + S^{-1}C^T E(v(t))), \\
\frac{d(E(\tilde{\theta}))}{dt} = -P(t)Y^T(t)C^T CE(\tilde{x}) + P(t)Y^T(t)C^T E(v(t)),
\end{cases}
\tag{18}
$$

where $E$ is an operator for calculating the mean value. As $w(t)$ and $v(t)$ are Gaussian distribution signals with zero mean, it can be derived that

$$
\begin{cases}
\frac{d(E(\tilde{x}))}{dt} = (A - S^{-1}C^T C)E(\tilde{x}) + E(\Phi(\hat{x}, x)) + E(\check{\Phi}_1(\hat{x}, x))\theta + \Phi_1(\hat{x}, u)E(\tilde{\theta}) + Y(t)E(\dot{\tilde{\theta}}), \\
\frac{d(E(\tilde{\theta}))}{dt} = -P(t)Y^T(t)C^T CE(\tilde{x}).
\end{cases}
\tag{19}
$$

Then we have $E(\tilde{x}) \to 0, E(\tilde{\theta}) \to 0$ when $t \to \infty$. The proof is completed. $\square$

According to Theorem 1, it is known that the mean value $E(\hat{\theta})$ of observed parameters $\hat{\theta}$ can converge to the true value of $\theta$. Therefore, the proposed detection algorithm is available for the fault detection of n-DoF robot manipulator.

## 4. Health Degree Assessment

Based on the fault detection algorithm proposed above, the observed parameters $\hat{\theta}$ can be obtained. Here, we take the mean value $\bar{\theta}(t) = \frac{1}{T}\int_{t-T}^{t}\hat{\theta}(\tau)d\tau$ of $\hat{\theta}$ over a period of time as a criterion for health assessment.

For the n-link (n-joint) robot manipulator, a weight vector $\omega = [\omega_1, \omega_1, \cdots, \omega_n]^T$ is given according to the importance of each joint. We define a nonlinear function

$$
f(x, a, b) = \begin{cases} 0, |x - a| < b, \\ g(|x - a| - b), otherwise, \end{cases}
\tag{20}
$$

with a convex function $g(\cdot)$, then the health degree of the robot manipulator is evaluated by $G(\bar{\theta}, \alpha, h) = h - \omega^T f(\bar{\theta}, 1, \alpha)$, where $h \in R^+$ is the upper bound of the health degree, $\alpha \in R^{+N}$ is the nonlinear dead zone threshold.

Once the health degree $G(\bar{\theta}, \alpha, h)$ drops below a certain threshold that can be set in accordance with the specific situation, relevant alarm and emergency measures should be implemented timely. Therefore, the health assessment index can be used to evaluate the failure risk of the robot manipulator.

## 5. Simulation

In the simulations, we consider a 2-DoF robot manipulator moving in the vertical plane with gravity. The robot links are assumed to be a rod of length 1 m and 0.8 m with concentrated mass at the rod end of 1 kg and 1.5 kg, respectively. The dynamics of the 2-DoF robot manipulator can be described as

$$
M(q) = \begin{bmatrix} (m_1 + m_2)l_1^2 + m_2l_2^2 + 2m_2l_1l_2\cos(q_2) & m_2l_2^2 + m_2l_1l_2\cos(q_2) \\ m_2l_2^2 + m_2l_1l_2\cos(q_2) & m_2l_2^2 \end{bmatrix},
\tag{21}
$$

$$
C(q, \dot{q}) = \begin{bmatrix} -m_2l_1l_2\dot{q}_2\sin(q_2) & -m_2l_1l_2(\dot{q}_1 + \dot{q}_2)\sin(q_2) \\ m_2l_1l_2\dot{q}_1\sin(q_2) & 0 \end{bmatrix},
\tag{22}
$$

$$
G(q) = \begin{bmatrix} (m_1 + m_2)gl_1\cos(q_2) + m_2gl_2\cos(q_1 + q_2) \\ m_2gl_2\cos(q_1 + q_2) \end{bmatrix},
\tag{23}
$$

where $m_1 = 1, m_2 = 1.5, l_1 = 1, l_2 = 0.8, w(t), v(t)$ are white Gaussian noise signals with mean 0 and variance 0.01.

We set $u(t) = [2,2]^T$, $k = 10$, $\hat{\theta}(0) = [1,1]^T$, $P(0) = I_2$, $Y(0) = [\ 0 \quad I_2\ ]^T$, $x(0) = [1,1,1,1]^T$ and $\hat{x}(0) = [0,0,0,0]^T$. The simulation results are presented in Figure 1.

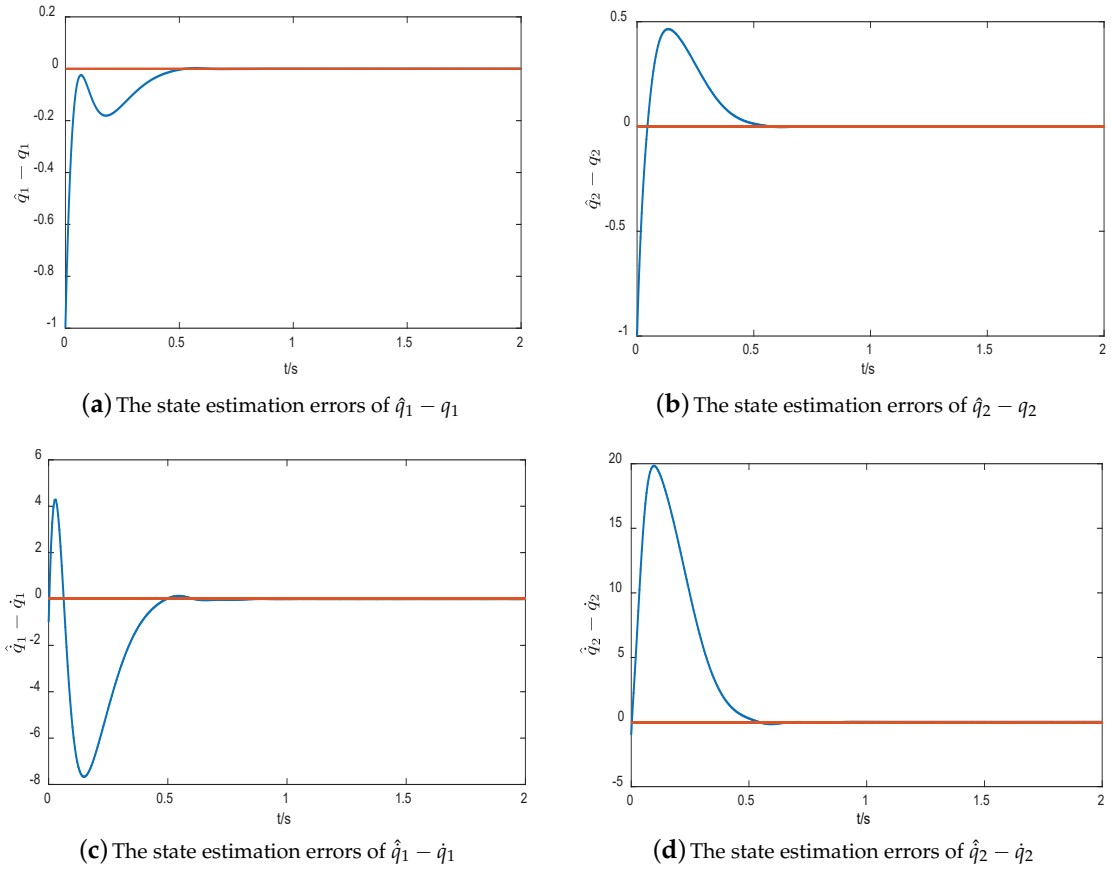

(**a**) The state estimation errors of $\hat{q}_1 - q_1$

(**b**) The state estimation errors of $\hat{q}_2 - q_2$

(**c**) The state estimation errors of $\hat{\dot{q}}_1 - \dot{q}_1$

(**d**) The state estimation errors of $\hat{\dot{q}}_2 - \dot{q}_2$

**Figure 1.** The state estimation errors of four states.

As discussed in Section 3, the actual dynamic robotic system can be regarded as a drive system, and our proposed ASO is treated as the response system. The errors between estimated state values and true values can be seen in Figure 1. The four states eventually converge to the real values, and the errors approach zero. This demonstrates the convergence of the proposed algorithm.

In Figure 2, we testify the performance of our algorithm for the observation of unknown parameters. For comparison, we introduce the extended state observer (ESO) proposed in [23]. Before 10th second, the real system runs at the expected state, i.e., $\theta = [1,1]$. At 10th second, the fault occurs due to several unpredictable external or internal disturbances, and $\theta = [0.6, 0.8]$. From Figure 2, it can be seen that due to the random fluctuations in the system, the estimated parameters based on both ASO and ESO will eventually converge to a value near the true value. However, the proposed ASO shows a faster convergence rate than the ESO. This also verifies the effectiveness of our fault detection algorithm.

Choose $T = 1$, we can obtain $\bar{\theta}_1$, $\bar{\theta}_2$ based on the health assessment approach proposed in Section 4. Compared with Figure 2, the mean value of the estimated parameters over a period of time is closer to the true value as shown in Figure 3. This is consistent with our theorem. In addition, this also shows that a health assessment $\bar{\theta}$ based on the mean estimation over a period of time is more effective than that based on only one estimation.

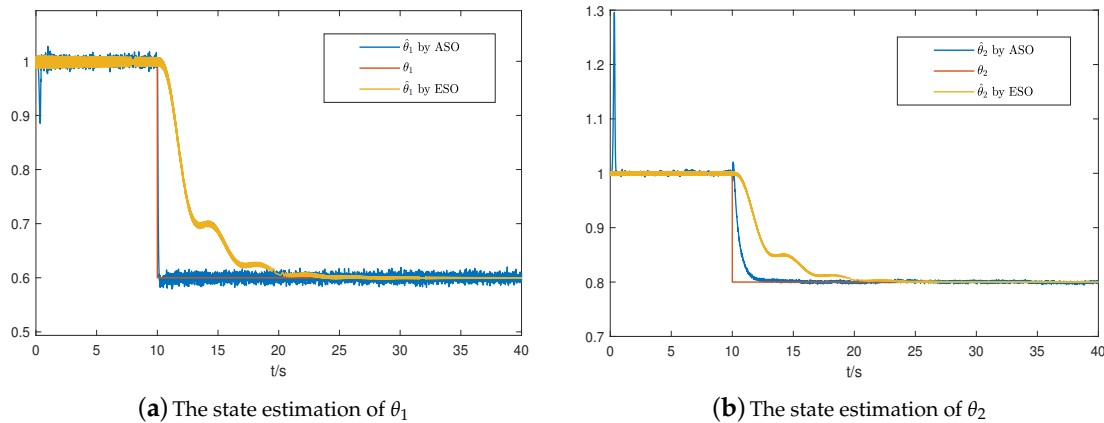

(**a**) The state estimation of $\theta_1$　　　　　　　(**b**) The state estimation of $\theta_2$

**Figure 2.** The state estimation of unknown parameters.

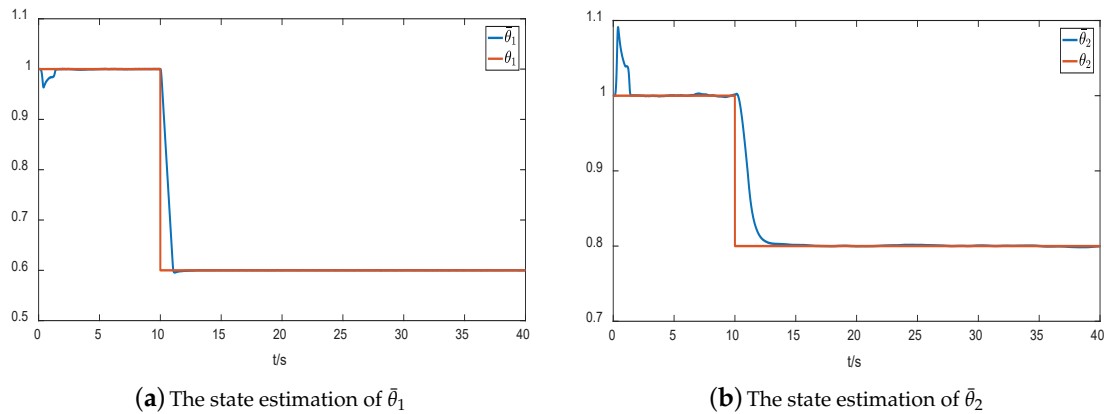

(**a**) The state estimation of $\bar{\theta}_1$　　　　　　　(**b**) The state estimation of $\bar{\theta}_2$

**Figure 3.** The state estimation of $\bar{\theta}_1$ and $\bar{\theta}_2$.

Finally, we select $\alpha = 0.05$, $h = 100$ and $\omega = [0.5, 0.5]^T$ to quantify the health degree. Given a nonlinear function $g(\cdot)$ as a quadratic function $g(x) = 500x^2$, the evolution curve of the health degree of the 2-DoF robot manipulator over time is shown in Figure 4. As the fault occurs, the health degree drops sharply to around 60, which verifies the rapidity of our fault detection algorithm and the applicability of our health degree assessment index.

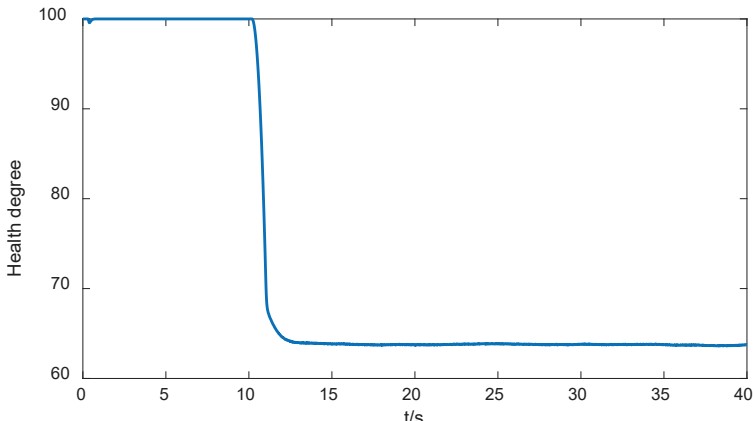

**Figure 4.** The health degree of the 2-DoF robot manipulator.

It is noted that the 2-DoF robot manipulator in the simulations is used only as an illustrative example for testifying the performance of the proposed fault detection algorithm and health degree

assessment index. Since our theoretical analysis in Sections 3 and 4 is based on the dynamic model of an n-DoF robot manipulator (1), the results of Figures 1–4 can be generalized to robot manipulators with arbitrary DoF.

## 6. Conclusions

In this paper, a state-space model of the n-link manipulator is formulated based on the physical characteristics of manipulators. The internal system disturbance and the output measurement disturbance are further considered. A novel fault detection algorithm for the robot manipulator is proposed. The algorithm employs an adaptive observer to reconstruct the real robot manipulator system, and compares the real system with the original system to judge the fault possibility. Furthermore, we propose a health assessment approach based on the proposed fault detection algorithm, which can better reflect the possibility of robot manipulator failure. The validity of our fault detection algorithm and the rationality of our health assessment index are verified through experimental data with a 2-degree-of-freedom robot manipulator. Due to the high penetration of robot equipments in the current industrial field, it is envisioned that our work will provide an effective method for the risk management in real systems. In the future, we will apply the proposed method to the automated production and inspection lines in the electrical industry.

**Author Contributions:** S.D. proposed the fault detection scheme. Z.K. draft the paper. L.P. wrote and reviewed the paper. Y.J. performed the simulation and optimized the models. Y.Z. analyzed the results and modified the paper. All authors have read and agreed to the published version of the manuscript.

**Funding:** This research was funded by the China Southern Power Grid Company Limited (GDKJXM20172756).

**Conflicts of Interest:** The authors declare no conflict of interest.

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
