# Peer review of "Adaptive State Observer for Robot Manipulators Diagnostics and Health Degree Assessment"

_applsci, doi:10.3390/app10020514_

Round 1
Reviewer 1 Report
I found two main issues in the present paper:
style ->
a) many sentences are nonsense (for instance, "Some commonly employed techniques in the literature, such as parameter estimation and state observer." on p.1 - I think some words are missing there. Moreover, on p. 2, "they presented a nonlinear to estimate the .." -- once more, some words are missing).
b) What does it mean the "fault degree"? Is it the fault entity? Or the fault size? Or? Please, also check what do you mean with "With the increment of errors, the seriousness of the robot manipulator fault increases" on p. 2.
c) What is a "link angular of the manipulator"? Please, correct.
d) Please check the typos
clarity of technical explanation ->
a) I'm unable to catch the novelty of the proposed approach: the authors claim to "consider the internal disturbance of the system and the output measurement disturbance" on p.2, and later they suppose to deal with robot joint actuator failures. So, what is the novelty? The proposal seems to deal with robot joint actuators (drives) faults, using noisy measurements, a well-known situation. Why there is no comparison with similar techniques, voted to detect robot joint actuators faults using noisy measurements? An example could enhance the novelty of the proposal, in my opinion.
b) the central technical part of the paper requires more b) the central technical part of the paper requires more clarity, in my opinion. In particular, Eq. 5 introduces various undefined components: what are P(t) and Y(t)? Please, explain in detail.
c) Add a comparison with other fault detection techniques, voted to solve the same task.
Author Response
Thanks very much for valuable comments and suggestions. According to these suggestions, we carefully revised our paper. Please see the attachment on our point-by-point responses.
We hope the current manuscript has been improved satisfactorily.

Reviewer 2 Report
The manuscript is well written and structured. It presents the modelling of an n-link manipulator. Considering internal disturbances of the system and measurement disturbances, the paper proposed a novel fault detection algorithm which employs an adaptive observer and validated with simulation results. The article is interesting although I have some minor comments which authors may consider addressing.
Although the introduction describes the gap in the field, the authors may consider analysing the cited papers critically rather than explaining the contributions of those. The authors may consider describing the 2DOF robot manipulator before stating the equation 21. A brief description of the system will be useful for the readers as no description of the simulation model is explained in the manuscript. As the work is validated against the simulation environment, the authors may consider explaining the future plan from this work and how to validate with n-dof (more than 2 dof) robot manipulators.I would recommend this article for publication after addressing the above-mentioned comments.
Author Response

(The authors gave the same response as above.)

Round 2
Reviewer 1 Report
No further remarks. Just correct the legend in Fig. 2b: in my opinion, the third label (corresponding to the yellow line) should be EOS instead of ASO.
Author Response
Thanks for your suggestion. We have corrected this typo in the legend in Fig. 2b.